# An Evaluation of Input–Output Value for Sustainability in a Chinese Steel Production System Based on Emergy Analysis

**Fengjiao Ma [1,2], A. Egrinya Eneji [3] and Yanbin Wu [1,2,*]**

[1] School of Management Science and Engineering, Hebei University of Economics and Business, Shijiazhuang 050061, Hebei, China; mafengjiao@heuet.edu.cn

[2] GIS Big Data Platform for Socio-Economy in Hebei, Shijiazhuang 050061, Hebei, China

[3] Department of Soil Science, Faculty of Agriculture, University of Calabar, Calabar PMB 1115, Nigeria; aeeneji@yahoo.co.uk

\* Correspondence: wuyanbin080@126.com; Tel./Fax: +86-311-87656207

**Abstract:** The social investment, natural resource consumption, and pollutant emissions involved in steel production can be evaluated comprehensively using the emergy analysis. We explored the sustainability of the steel production system from four aspects: input index, output index, input–output index, and sustainability index. The results showed that the maximum inputs were the intermediate product/recyclable materials produced within the production line; energy sources were mainly non-renewable and the emergy value of pollutants discharged was rather low. The environmental load rate of the pelletizing and sintering processes were the highest and the proportion of recycled materials for puddling and steel-making were the highest. The emergy investment rate of rolling was the highest; the emergy value of the pollutants discharged in each process was very small, and the emergy yield ratio was highest in the rolling process. Pelletizing, sintering, and steel-making were input consuming processes, but the sustainability index of puddling and rolling processes was sound. The whole process line can be sustainable, considering the useful intermediate and recyclable products.

**Keywords:** emergy analysis; pollution impact; resource consumption; steel production; sustainable development

## 1. Introduction

Steel is widely used in construction, transportation, packaging, renewable energy, and other industries and the world's crude steel output exceeded 1.6 billion tons in 2016 [1]. However, it is also an energy-intensive industry, whose carbon dioxide emissions account for 6% to 7% of global anthropogenic carbon dioxide emissions due to large amounts of fossil fuel consumption [2]. The treatment of solid waste such as steel slag, iron dust, and coal ash generated during production has caused a series of environmental problems [3]. Steel production relies on the natural ecosystem and human economic system feedback resources and the resulting waste flows into the natural ecosystem and could affect human health. A research framework that considers the human economic system, natural ecosystem, and the steel production system is required to evaluate the sustainable development of the steel industry. The ecological economics evaluation method that comprehensively considers economic development, resource consumption and environmental protection is an important tool for evaluating sustainable development. Its application to the steel industry is an important research topic for the sustainable management of the industry.

Among the existing eco-economic evaluation methods, the material flow analysis does not consider the contribution of the ecosystem to production [4,5]; the evaluation using life cycle assessment is based on human preferences [6]; economic analysis mainly depends on market and shadow prices, and its outcome is not objective enough; energy analysis usually does not consider the different effects provided by energy from different sources [7,8].

In contrast to other analytical methods, H.T. Odum considered the natural energy hierarchy of the universe in which many joules of one kind must be degraded to generate a few joules of another and propose the concept of "emergy" [9]. Odum measures, values, and aggregates energy of different types by their transformities. Transformities, defined as the emergy per unit energy, are calculated as the amount of one type of energy required to produce a heat equivalent of another type of energy. To account for the difference in quality of thermal equivalents among different energies, all energy costs are measured in solar emjoules (sej), the quantity of solar energy used to produce another type of energy. Fuels and materials with higher transformities require larger amounts of sunlight to produce and therefore are considered more economically useful [10]. The emergy analysis is an energy ecological method based on the principle of physical thermodynamics. The indicators of economic system and ecosystem can be uniformly converted into emergy values. By incorporating aspects of energy quality and ecological hierarchy to evaluate the contribution of the natural environment to the human-economic system, this methodology allows for balancing of the needs of both human and natural systems, expressing the socio-economic-environmental effects in common terms [11]. Emergy with corresponding indices and ratios has been proved to be an effective and robust tool to understand the resource flows supporting both the natural ecosystem and macro-economic system, and can be used to measure their overall performances and sometimes sustainability [12]. This method has been widely accepted as an effective ecological evaluation tool to assess comprehensive performances of all kinds of systems with different scales and functions [13–16].

In the field of industrial production, Brown and Ulgiati added ecological service indicators to the emergy production system to evaluate the power production system [17]. Geng et al. used emergy analysis to evaluate the environmental performance and sustainability of industrial parks [18] and Yuan et al. analyzed the recycling effects of different methods for construction waste through the emergy theory [19]. In the field of renewable energies industry, a comprehensive energy and economic assessment of biofuels was conducted by Ulgiati, based on economy, energy, and emergy and a proposal to integrate ethanol production with industrial activities with a "zero emission framework" was suggested [20]. Takahashi and Ortega made an emergy assessment of oleaginous crops cultivated in Brazil, available to produce biodiesel, to determine which crop is the most sustainable [21]. Zhou et al. analyzed a farm biogas based on emergy analysis and found that the farm biogas project has more reliant on the local renewable resources input, less environmental pressure and higher sustainability compared with other typical agricultural systems [22]. In the field of steel production, Zhang et al. used emergy analysis to assess the sustainability of Chinese steel production from 1998 to 2008, showing that its sustainability was very low and continued to decline [23]. Pan et al. evaluated the sustainability of Chinese steel eco-industrial parks based on the emergy theory and found that after the implementation of material recycling and energy cascade utilization, all indicators were superior to the traditional production chain [24].

In order to understand the energy efficiency, environmental impact, and sustainable development of steel industry, a systematic method to measure the comprehensive performances of steel enterprise is urgent. The emergy analysis can be an effective method for evaluating sustainable development, considering the social investment, natural resource consumption, and impacts of pollutant emission from the steel industry. However, the current application of emergy analysis to the steel industry has only focused on the sustainable development from a fixed resource type. A detailed inquiry into the various material resources for the steel production process is needed to analyze the productivity and sustainability of the steel industry. Therefore, we explored the detailed inputs of renewable and non-renewable resources from three aspects: natural ecosystem, human economic system, and steel

production system. In addition, we analyzed each sub-link of the steel production line to explore the status and potential of energy consumption. Finally, the efficiency and sustainable development of steel production were examined in detail from the input-, output-, input–output- and comprehensive sustainability indexes of steel production. This will allow for the examination of the dependence of steel production on different systems as well as the role of recycling in the production process and identification of the sustainable development index that considers the environmental impacts and waste discharge.

## 2. Materials and Methodology

### 2.1. Evaluation Framework

Our research target was a steel production system (Figure 1), whose boundary is the area of steel production enterprise. Steel production consumes a lot of resources, and generates various wastes. Three categories of system are defined for emergy accounting and for the understanding of the system interactions. (1) The natural system represents the natural environment, which has not been substantially altered by human intervention. (2) The human economics system is dominated by human beings and deals with the production, distribution, and consumption of goods and services in a particular society. (3) The steel production system in this paper refers to an industrial system that contains pelletizing, sintering, puddling, steel-making, rolling, and other related auxiliary process. The resources are derived from natural and human economic systems; the products are sold to the human economic systems, the pollutants are returned to the natural ecosystem while affecting human health, and some wastes that could be reused (here defined as recyclable materials) are returned to the production line. Based on the emergy algorithm, we abbreviate the renewable resources from natural system as R, the non-renewable resources from natural system as N, the renewable resources from human economics system as $F_R$, the non-renewable resources from human economics system as $F_N$ and the product for human economic system as Y. In addition, some products (such as sinter, pellet, etc.) are defined as intermediate products, because they can be sold on the market, but they are also used in other parts of the steel production system. However, the effect of pollutant emissions from the steel production plant on other systems is useless or even harmful. Here, we used dotted lines to describe their pathway (Figure 1). The production process could refer to the entire steel production line, but also to a sub-process, such as sintering process.

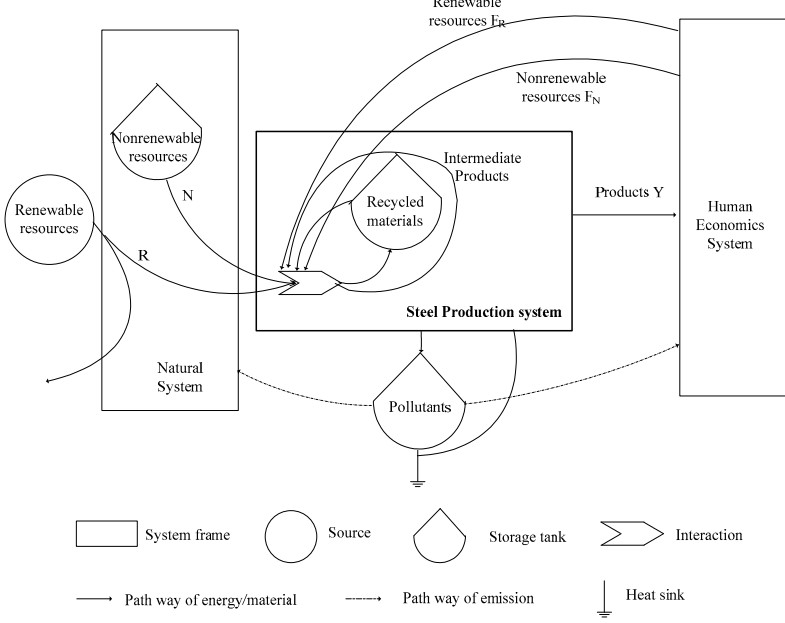

**Figure 1.** Material and energy flow diagram of the steel production process.

The inputs of different systems include renewable and non-renewable resources. The resource input from the human economic system is also needed, to be supported by the natural ecosystem. For example, the electricity supplied by the human economic system depends on coal or water resources supplied by the natural system. However, considering that the power system needs a large number of other production equipment, the proportion of natural resources input is relatively low, so the electricity is classified into the resource input of the human economic system. The natural resources, such as coal and lime, are direct supplies of the natural ecosystem.

### 2.2. Data Collection and Calculation of Emergy

#### 2.2.1. Data Collection

This study explored the sustainable development of steel production system. The steel factory studied was a combined factory, consisting of sintering, pelletizing, puddling, steel making, steel rolling and power generation. It had an annual production capacity of 1.2 million tons of pellets, 9.15 million tons of sinter, 4.65 million tons of pig iron, 4.5 million tons of billets and 3.2 million tons of coils. Its production pathway is shown in Figure 2. The products of the factory can enter the market or the next production link of the factory directly.

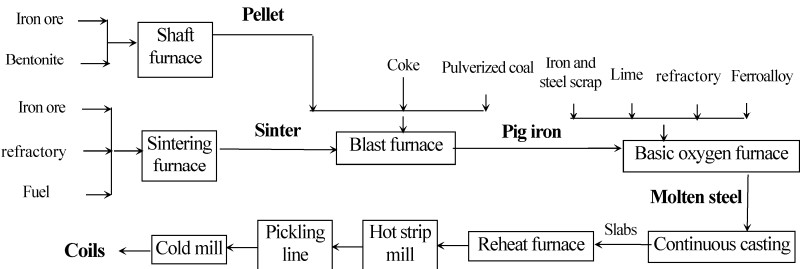

**Figure 2.** Flow diagram for the steel production process.

Data Source: Considering the different conditions of production across years and the imperfection of some material flow monitoring, we used data from the environmental impact assessment report of a standard steel factory—the Yuhua Steel Co. Ltd. in Wuan City, China. We collected the report directly from the authors, who conducted field investigation and technical demonstration on the entire steel enterprise. Data were collected for natural renewable and non-renewable resources, human economic renewable and non-renewable resources, intermediate products and recycled materials as the input raw data; pollutants, products, intermediate products, and recycled materials were the output raw data.

Data processing methods: Various input–output indexes must be comparable to evaluate the efficiency and sustainability of the whole system. In this study, the emergy analysis method was used. The specific algorithm was firstly to convert different input and output elements into energy or mass data, followed by calculation of the emergy conversion rate. Finally, the original data were multiplied by the emergy conversion rate to obtain the solar emergy value (sej) of each index (Table 1).

#### 2.2.2. Impact Evaluation of Emissions

For the steel production process, although most input–output indexes could be calculated as the product of original data and emergy conversion rate, the pollutants produced in the process could not be simply multiplied by their emergy conversion rates. Because pollutants are harmful to people and environment rather than a useful resource, their emergy value should be calculated from their negative effects.

Even if the pollutants from the production process are within the national permissible limits after remediation, there are still some gaps between the emission concentration and the environmental quality standards suitable for human survival. These pollutants need a lot of air within the environment

to dilute to the acceptable concentration. This environmental service is defined as services for diluting pollutants. However, these pollutants may cause ecological and economic losses (biodiversity loss, ecosystem degradation, damage to human health) before reaching the acceptable concentration. These losses contain certain emergy values. Therefore, the ecological impacts of effluent pollutants are in two parts: dilution of ecosystem services and emergy loss of emission.

The calculation of emergy for diluting ecological services was done according to Ulgiati and Brown [25]. However, the pollutants are regarded as by-products in the literature [25] and the service used for dilution is regarded as renewable resources provided by the environment. We considered the pollutants as harmful waste rather than by-products since they cannot be utilized under the current technical level of the research enterprise. As an effluent waste, pollutants can only be a harmful substance, whose disadvantages are expressed by the damage to natural and human resources. The value of this damage is essentially negative emergy.

① Calculation of Emergy for Diluting Ecological Services

Firstly, the environmental quality of diluted pollutants was calculated. The pollutant studied in this paper was only exhaust gas. Therefore, the air quality of diluted exhaust gas was calculated as

$$M_{d,air} = d \times \frac{W}{c} - M_{e,air} \tag{1}$$

where $M_{d,air}$ is the quality of the air used to dilute pollutants; $M_{e,air}$ is the quality of the air emitted from the steel production process; $d$ is the air density (1.23 kg/m$^3$); $W$ is the amount of pollutants discharged annually; $c$ is the acceptable concentration of pollutants.

The mass of diluted air was converted to emergy by diluting the kinetic energy of air. By multiplying by the conversion rate of emergy, the emergy value of diluted air can be obtained.

$$Em_{d,air} = E_{d,air} \times Tr_{air} = \frac{1}{2} \times M_{d,air} \times v^2 \times Tr_{air} \tag{2}$$

where $Em_{d,air}$ is the emergy value of dilute air; $E_{d,air}$ is the kinetic energy of dilute air; $Tr_{air}$ is the transformity of wind, here it is $1.50 \times 10^3$ sej/J; $M_{d,air}$ is the quality of the air used to dilute pollutants; $v$ is average annual wind speed, here it was chosen as 1.5 m/s.

② Calculation of emergy loss of emission

Human resources were considered as a slow renewable resource, and the generation and use of pollutants would lead to irreversible losses. In this report, the *DALY* method proposed by WHO was used to quantitatively assess the damage of pollutants to human beings [22]. Emergy loss was calculated as

$$Em_{manpower} = \sum M_i \times DALY_i \times \tau_H \tag{3}$$

where $Em_{manpower}$ is the emergy of human resource loss, sej; $i$ is the $i$th pollutant; $M_i$ is the quality of the $i$th pollutant; $DALY$ is the impact factor of the $i$th pollutant; $\tau_H$ is the emergy unit of human resources per year, which is equal to the annual total emergy use of a country or region divided by its population, and here $\tau_H$ equaled $1.32 \times 10^{16}$ sej/person [24,26].

The specific indicators and results of the calculation are shown in Table 1.

2.2.3. Emergy Evaluation of the Steel Industry

From the material and energy flow diagram of the production process, the different sub-processes of the steel factory were systematically sorted out. The input and output indexes were converted into heat or mass data. The original data were multiplied by the corresponding emergy conversion rate to obtain the solar emjoule value (sej) of each index. Because of the large amount of data, the summary results are shown in Table A1 at the end of this paper. The main body of this paper only gave the emergy input–output statistics of the steel production system (Table 2).

**Table 1.** Yearly emergy estimate of pollutants in the steel production process.

| Production Process | Pollutant | Pollutant Discharge Mass/t | Discharge Volume of Waste Gas/m$^3$ | Acceptable Concentration/µg/m$^3$ [27] | Mass of the Air Used to Dilute Pollutant/t | Emergy of the Air Used to Dilute Pollutant/sej | DALY/Kg of Emission | Emergy Loss of Emission/sej | Emergy of Total Impacts of Pollutants/sej |
|---|---|---|---|---|---|---|---|---|---|
| Pelletizing | SO$_2$ | 295.26 | $3.11 \times 10^9$ | 20 | $1.82 \times 10^{10}$ | $3.06 \times 10^{16}$ | $5.46 \times 10^{-5}$ [28] | $2.13 \times 10^{17}$ | $2.43 \times 10^{17}$ |
| | Dust | 85.51 | $5.10 \times 10^9$ | 80 | $1.31 \times 10^9$ | $2.21 \times 10^{15}$ | $3.75 \times 10^{-4}$ [28] | $4.23 \times 10^{17}$ | $4.25 \times 10^{17}$ |
| | NO$_x$ | 530.91 | $3.12 \times 10^9$ | 50 | $1.31 \times 10^{10}$ | $2.20 \times 10^{16}$ | $8.87 \times 10^{-5}$ [29] | $6.22 \times 10^{17}$ | $6.44 \times 10^{17}$ |
| Sintering | SO$_2$ | 2315.38 | $1.99 \times 10^{10}$ | 20 | $1.42 \times 10^{11}$ | $2.40 \times 10^{17}$ | $5.46 \times 10^{-5}$ [28] | $1.67 \times 10^{18}$ | $1.91 \times 10^{18}$ |
| | Dust | 967.46 | $4.23 \times 10^{10}$ | 80 | $1.48 \times 10^{10}$ | $2.50 \times 10^{16}$ | $3.75 \times 10^{-4}$ [28] | $4.79 \times 10^{18}$ | $4.81 \times 10^{18}$ |
| | NO$_x$ | 4503.19 | $2.02 \times 10^{10}$ | 50 | $1.11 \times 10^{11}$ | $1.87 \times 10^{17}$ | $8.87 \times 10^{-5}$ [29] | $5.27 \times 10^{18}$ | $5.46 \times 10^{18}$ |
| Puddling | SO$_2$ | 178.81 | $3.85 \times 10^9$ | 20 | $1.10 \times 10^{10}$ | $1.85 \times 10^{16}$ | $5.46 \times 10^{-5}$ [28] | $1.29 \times 10^{17}$ | $1.47 \times 10^{17}$ |
| | Dust | 713.47 | $3.54 \times 10^{10}$ | 80 | $1.09 \times 10^{10}$ | $1.84 \times 10^{16}$ | $3.75 \times 10^{-4}$ [28] | $3.53 \times 10^{18}$ | $3.55 \times 10^{18}$ |
| | NO$_x$ | 449.03 | $3.85 \times 10^9$ | 50 | $1.00 \times 10^{10}$ | $1.86 \times 10^{16}$ | $8.87 \times 10^{-5}$ [29] | $5.26 \times 10^{17}$ | $5.44 \times 10^{17}$ |
| Steel-making | Dust | 528.30 | $2.24 \times 10^{10}$ | 80 | $8.11 \times 10^9$ | $1.37 \times 10^{16}$ | $3.75 \times 10^{-4}$ [28] | $2.62 \times 10^{18}$ | $2.63 \times 10^{18}$ |
| Rolling | SO$_2$ | 84.33 | $2.00 \times 10^9$ | 20 | $5.18 \times 10^9$ | $8.75 \times 10^{15}$ | $5.46 \times 10^{-5}$ [28] | $6.08 \times 10^{16}$ | $6.95 \times 10^{16}$ |
| | Dust | 37.85 | $2.00 \times 10^9$ | 80 | $5.80 \times 10^8$ | $9.78 \times 10^{14}$ | $3.75 \times 10^{-4}$ [28] | $1.87 \times 10^{17}$ | $1.88 \times 10^{17}$ |
| | NO$_x$ | 170.97 | $2.00 \times 10^9$ | 50 | $4.20 \times 10^9$ | $7.09 \times 10^{15}$ | $8.87 \times 10^{-5}$ [29] | $2.00 \times 10^{17}$ | $2.07 \times 10^{17}$ |
| Power Plant | SO$_2$ | 535.52 | $1.08 \times 10^{10}$ | 20 | $3.29 \times 10^{10}$ | $5.56 \times 10^{16}$ | $5.46 \times 10^{-5}$ [28] | $3.86 \times 10^{17}$ | $4.42 \times 10^{17}$ |
| | Dust | 87.18 | $1.08 \times 10^{10}$ | 80 | $1.33 \times 10^9$ | $2.24 \times 10^{15}$ | $3.75 \times 10^{-4}$ [28] | $4.32 \times 10^{17}$ | $4.34 \times 10^{17}$ |
| | NO$_x$ | 424.64 | $1.08 \times 10^{10}$ | 50 | $1.04 \times 10^{10}$ | $1.76 \times 10^{16}$ | $8.87 \times 10^{-5}$ [29] | $4.97 \times 10^{17}$ | $5.15 \times 10^{17}$ |

**Table 2.** Emergy input and output in the steel production system.

**Input**

| Items | Resource Type | Indexes | Emergy sej |
|---|---|---|---|
| Natural system | Renewable resources (R) | Fresh water | $2.35 \times 10^{18}$ |
|  |  | Air | $2.30 \times 10^{20}$ |
|  | Non-renewable resources (N) | Bentonite | $1.92 \times 10^{19}$ |
|  |  | Powdered iron | $7.58 \times 10^{21}$ |
|  |  | Limestone | $1.46 \times 10^{21}$ |
|  |  | High magnesium powder | $8.59 \times 10^{19}$ |
|  |  | Iron ore | $8.70 \times 10^{19}$ |
|  |  | Coal | $2.02 \times 10^{20}$ |
|  |  | Pulverized coal | $2.62 \times 10^{19}$ |
|  |  | Iron block | $9.41 \times 10^{19}$ |
|  |  | Ferroalloy | $1.14 \times 10^{19}$ |
|  |  | Doomite | $6.55 \times 10^{19}$ |
|  |  | Flour | $6.00 \times 10^{17}$ |
|  |  | Soil loss | $7.53 \times 10^{20}$ |
| Human economics system | Renewable resources ($F_R$) | Labor | $2.64 \times 10^{20}$ |
|  |  | Investment in fixed assets | $2.32 \times 10^{20}$ |
|  | Non-renewable resources ($F_N$) | Thermal power electricity | $6.78 \times 10^{20}$ |
|  |  | Coke powder | $4.15 \times 10^{9}$ |
|  |  | Coke | $4.71 \times 10^{20}$ |
|  |  | Nut coke | $7.76 \times 10^{18}$ |
|  |  | White ash | $1.01 \times 10^{21}$ |
| Steel production system | Intermediate products | Sinter | $7.23 \times 10^{21}$ |
|  |  | Pellet | $1.31 \times 10^{21}$ |
|  |  | Pig iron | $1.01 \times 10^{22}$ |
|  |  | Billet steel | $1.001 \times 10^{22}$ |
|  | Recycled materials | Blast furnace gas | $2.121 \times 10^{21}$ |
|  |  | Convertor gas | $2.87 \times 10^{20}$ |
|  |  | Dust and ash | $4.00 \times 10^{19}$ |
|  |  | Water treatment sludge | $5.39 \times 10^{19}$ |
|  |  | Sinter reentry | $2.01 \times 10^{20}$ |
|  |  | Pellet return | $1.09 \times 10^{20}$ |
|  |  | Steam consumption | $5.74 \times 10^{19}$ |
|  |  | Steel scrap | $9.89 \times 10^{19}$ |
|  |  | Electricity | $3.03 \times 10^{20}$ |
|  |  | Nitrogen | $9.03 \times 10^{20}$ |
|  |  | Oxygen | $1.38 \times 10^{20}$ |

**Output**

| Items | Resource Type | Indexes | Emergy sej |
|---|---|---|---|
| Natural system | Pollutants | SO$_2$ | $2.81 \times 10^{18}$ |
|  |  | Dust | $1.20 \times 10^{19}$ |
|  |  | NO$_x$ | $7.37 \times 10^{18}$ |
| Human economics system | Products (Y) | Sinter | $2.75 \times 10^{21}$ |
|  |  | Pig iron | $2.51 \times 10^{21}$ |
|  |  | Billet steel | $3.88 \times 10^{21}$ |
|  |  | Rolled steel | $9.89 \times 10^{21}$ |
| Steel production system | Intermediate products | Pellet | $1.31 \times 10^{21}$ |
|  |  | Sinter | $7.23 \times 10^{21}$ |
|  |  | Pig iron | $1.01 \times 10^{22}$ |
|  |  | Billet steel | $1.00 \times 10^{22}$ |
|  | Recycled materials | Desulphurizing Slag | $6.20 \times 10^{18}$ |
|  |  | Dust and ash | $1.57 \times 10^{20}$ |
|  |  | Desulphurized gypsum | $2.36 \times 10^{19}$ |
|  |  | Sinter reentry | $8.26 \times 10^{20}$ |
|  |  | Blast furnace slag | $1.21 \times 10^{21}$ |
|  |  | Blast furnace gas | $2.12 \times 10^{21}$ |
|  |  | Hot blast stove flue gas | $2.29 \times 10^{22}$ |
|  |  | Steel slag | $1.32 \times 10^{21}$ |
|  |  | Dust mud | $5.38 \times 10^{19}$ |
|  |  | Iron oxide skin | $2.38 \times 10^{19}$ |
|  |  | Remainder residue | $1.62 \times 10^{19}$ |
|  |  | Steam production | $5.58 \times 10^{19}$ |
|  |  | Converter gas production | $2.87 \times 10^{20}$ |
|  |  | Iron oxide sludge | $2.19 \times 10^{19}$ |
|  |  | Steel scrap | $8.75 \times 10^{19}$ |
|  |  | Electricity | $3.03 \times 10^{20}$ |
|  |  | Nitrogen | $9.03 \times 10^{20}$ |
|  |  | Oxygen | $1.38 \times 10^{20}$ |

### 2.2.4. Emergy Indexes Used in This Study

The emergy evaluation indexes were compiled according to the input–output system and resource utilization of the steel industry (Table 3).

**Table 3.** Emergy indexes used in steel production system.

| Items | Indexes | Formulation |
|---|---|---|
| Input index | Environment loading ratio (ELR) | Non-renewable resources $(N + F_N)$/Renewable resources $(R + F_R)$ |
| | Proportion of recycled materials used (PRM) | Recycled materials used/Total input |
| | Emergy investment ratio (EIR) | Human economics system input/Natural system input |
| Output index | Environmental impact rate (EnIR) | Pollutants/Products |
| | Product rate (PR) | Products/Total output |
| Input–output index | Emergy yield ratio (EYR) | Products/Human economics system input |
| | Total emergy yield ratio (TEYR) | (Products + Intermediate products + Recycled materials)/(Human economics system input + Natural system input) |
| | Net emergy yield ratio (NEYR) | (Products + Intermediate products + Recycled materials-Pollutants)/(Human economics system input + Natural system input) |
| | Emergy input–output rate (EIOR) | Products/Total input |
| | Total emergy input–output ratio (TEIOR) | (Products + Intermediate products + Recycled materials)/Total input |
| | Net emergy input–output ratio (NEIOR) | (Products + Intermediate products + Recycled materials − Pollutants)/Total input |
| Sustainability index | Emergy sustainable development index (ESDI) | Emergy yield ratio (EYR)/Environment loading ratio (ELR) |
| | Total emergy sustainable development index (TESDI) | Total emergy yield ratio (TEYR)/Environment loading ratio (ELR) |
| | Net emergy sustainable development index (NESDI) | Net emergy yield ratio (NEYR)/Environment loading ratio (ELR) |

## 3. Results and Discussion

### 3.1. Structure of Inputs and Outputs in the Steel Industry

The whole process of steel production was analyzed in terms of input and output. The beneficial effect of steel production on people was positive, and both its harmful effect on people and the use of human beneficial emergy were negative, as shown in Figure 3. There was little difference between input and output of the system; the emergy loss was minimal, and the emergy output rate was high.

Among the three systems, the steel production system had the highest overall input and output, which included intermediate and recyclable products. Apart from the intermediate products that could be sold and used directly, the proportion of recyclable materials in various input–output indicators was also the largest. In addition to gas and other resources, these materials were mostly solid wastes such as steel slag, dust particles, etc. After being treated and collected, they accounted for 43% of the emergy value of inputs. The harmless treatment of steel production process played an important role. If these materials were not properly recycled, more resources would need to be invested from the natural ecosystem and human economic system, and the impact of the associated direct emissions would be close to the beneficial emergy value derived from the product itself. Regardless of the input of steel production system, the input resources were mainly non-renewable resources, which was consistent with previous findings [23,24].

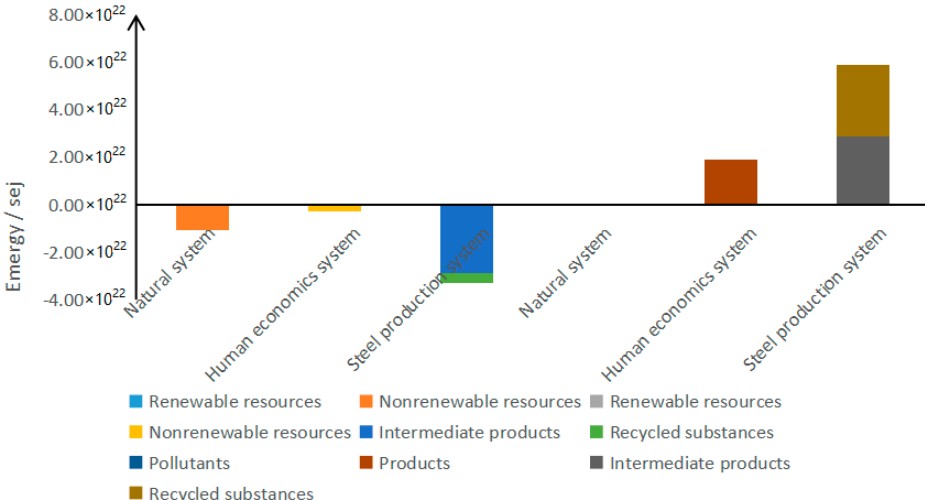

**Figure 3.** Structure of the emergy inputs and outputs in the steel production process.

In the input and output of each system, both renewable resources of natural ecosystem and human economic system accounted for a very low proportion, suggesting that the environmental load of steel production was high. In addition, the emergy value of pollutants discharged into the natural ecosystem was very low. Although the steel industry is a typical high pollution enterprise, the proportion of its pollution impact on overall input and output was not particularly serious based on the emergy analysis. Under the pressure of environmental protection, the steel factories have fared better. The impact of pollutants was relatively small when the pollutants were treated and reusable resources/wastes recovered as much as possible. In addition, compared with other steel enterprises in China [23,24], this research enterprise does not discharge waste water and the amount of waste gas pollutants was also significantly less. This indicated that the environmental protection technology of this enterprise was at the forefront in China.

### 3.2. Variation in Emergy of Different Production Links

Input–output indexes were analyzed from four aspects: input indexes, output indexes, the relationship between input–output indexes, and comprehensive sustainability indexes, considering different types of superimposed effects. The emergy production efficiency of each sub-production process and the whole production process was also analyzed (Table 4).

**Table 4.** Emergy indexes used in steel production system.

| Items | Indexes | Pelletizing | Sintering | Puddling | Steel-Making | Rolling | Whole Process |
|---|---|---|---|---|---|---|---|
| Input index | Environment loading ratio (ELR) | 31,191.381 | 145,338.975 | 14.418 | 21.239 | 39.481 | 17.242 |
| | Proportion of recycled materials used (PRM) | 0.014 | 0.061 | 0.090 | 0.098 | 0.024 | 0.093 |
| | Emergy investment ratio (EIR) | 0.177 | 0.038 | 1.588 | 4.876 | 39.481 | 0.251 |
| Output index | Environmental impact rate (EnIR) | 0.001 | 0.001 | 0.000 | 0.000 | 0.000 | 0.001 |
| | Product rate (PR) | 0.994 | 0.914 | 0.324 | 0.891 | 0.989 | 0.243 |
| Input–output index | Emergy yield ratio (EYR) | 6.987 | 32.669 | 20.920 | 12.234 | 42.917 | 8.777 |
| | Total emergy yield ratio (TEYR) | 7.020 | 35.708 | 64.647 | 13.735 | 43.397 | 21.992 |
| | Net emergy yield ratio (NEYR) | 7.013 | 35.668 | 64.640 | 13.732 | 43.395 | 21.982 |
| | Emergy input–output rate (EIOR) | 1.038 | 1.116 | 1.205 | 1.095 | 0.940 | 0.411 |
| | Total emergy input–output ratio (TEIOR) | 1.043 | 1.219 | 3.723 | 1.229 | 0.951 | 1.031 |
| | Net emergy input–output ratio (NEIOR) | 1.041 | 1.218 | 3.723 | 1.229 | 0.951 | 1.030 |
| Sustainability index | Emergy sustainable development index (ESDI) | 0.000 | 0.000 | 1.451 | 0.576 | 1.087 | 0.509 |
| | Total emergy sustainable development index (TESDI) | 0.000 | 587.384 | 4.484 | 0.647 | 1.099 | 1.276 |
| | Net emergy sustainable development index (NESDI) | 0.000 | 945.265 | 4.483 | 0.647 | 1.099 | 1.275 |

### 3.2.1. Input Indexes

By analyzing the relationship among different input indexes, the dependence of the production process on different systems and resource types could be understood. Environmental load rate (ELR) reflects the proportion of input of non-renewable and renewable resources. The ELR for pelletizing and sintering processes at the front end of production chain were much higher than other processes. Thus, the whole production process needed to invest a large amount of non-renewable resources to start production, then the demand for non-renewable resources was greatly reduced. The ELR of whole process was 17.242, which was lower than the values for other steel enterprises in China, but there was still some great environmental stress (ELR > 10) [23,24].

The proportion of recycled materials used (PRM) reflects the extent of waste disposal during steel production. If the waste materials cannot be recycled for further production, they can easily become an additional environmental burden. Therefore, the PRM index, acting much like the decomposer in the ecosystem, plays an important role in a sustainable industrial production system. Both puddling and steel-making processes have a high PRM, so that these processes can better absorb and digest the waste in the whole steel production process (Table 4).

Emergy investment ratio (EIR) is different from ELR and is used to explore the relationship between inputs from the human economic system and natural ecosystem. The EIR of the rolling process was higher because it had no input from the human economic system, except some electricity; so the proportion of natural resources input was significantly increased. In pelletizing, and sintering processes as well as the entire line, the input from the human economic system was much greater than that from the natural ecosystem, indicating that the dependence of steel production on human

economic system was greater than on natural ecosystem. The EIR of the entire line was similar to other steel enterprises, though it varied with years [23].

### 3.2.2. Output Indexes

Environmental impact rate (EnIR) reflects the emergy of discharged pollutants per unit product, which can measure the negative environmental impact of products. The EnIR of each process was low, and the EnIR of puddling, steelmaking, and rolling, were less than 0.001 (Table 4). Thus, the environmental costs of production consistent with emission standards was relatively small, based on emergy analysis. In particular, this paper considered pollutants as emissions and were analyzed among the output emergy; however, some studies regarded pollution as the loss of input emergy [24,25].

Product rate (PR) refers to the proportion of target products relative to all other outputs. The very low PR for the whole process was due to the fact that only the final steel was used as the product in this analysis, excluding the huge intermediate products and recycling materials. The PR of the puddling process was much lower than that of other sub-processes. Many of the emergy invested in the puddling process was converted into recycled materials which are subject to further processing. The emergy efficiency of the production cycle could be greatly increased by increasing the PR of the puddling process.

### 3.2.3. Input–Output Indexes

The traditional Emergy yield ratio (EYR) reflects the emergy of the output (product) under a certain amount of purchased emergy. It can be seen from the Table 4 that the EYR of rolling processes was much higher than that of other processes. The main reason was that compared with pelletizing and sintering processes, the most important resource inputs for rolling were the intermediate products produced in the previous process, which needed not be purchased. The EYR of steelmaking was low due to the high input of natural resources such as oxygen and nitrogen. The EYR of whole process was 8.777, which was larger than other steel enterprises, meaning the research enterprise was more competitive [23–25]. Total emergy yield ratio (TEYR) represents the output of all products (including the sum of final products and recyclable materials) under a certain purchased emergy. The net emergy yield ratio (NEYR) represents the output of all products minus pollutants under a certain purchased emergy. Because the emergy value of pollutants was much lower than other outputs, there was little difference between the TEYR and the NEYR. The difference of TEYR and NEYR from each process was similar to the difference of the EYR. The TEYR and NEYR of puddling process were high, because it used less purchasable resources. Also, not only the pig iron products, but the recyclable materials were produced with a great deal of emergy value. So, the investment rate of puddling process was much higher than that of other processes.

The emergy input–output rate (EIOR) is developed to reflect the amount of emergy products produced by the system, considering all the input resources at the same time. The total emergy input–output ratio (TEIOR) explores the total output of products, intermediate products and recyclables that are produced after inputting resources. The net emergy input–output ratio (NEIOR) represents the total product mentioned above minus the pollutant emergy value after inputting resources. Compared with the EYR, the EIOR not only considers the input and output efficiency of purchased resources, but also comprehensively analyzes the conversion efficiency of all input resources in the process. As shown in Table 4, the EIOR differed little among the various processes and the input–output ratio of each process was almost slightly greater than 1, and the emergy efficiency of each process was high. The TEIOR and the NEIOR of the puddling process were greater than 3. The emergy production and conversion efficiency was rather high under the comprehensive consideration of various inputs and outputs.

3.2.4. Sustainability Indexes

The emergy sustainable development index (ESDI) reflects the sustainable development of the system. The relationship between the ESDI and sustainable development can be summed up as follows: when the ESDI is greater than 1 but less than 10, it indicates that the system is developing and relatively dynamic, and the emergy of sustainable development is in good condition; an ESDI greater than 10 indicates that the economy is underdeveloped; ESDI less than 1 indicates a consumption-oriented system and the development is unsustainable [30]. The ESDI of pelletizing, sintering, puddling, steel-making, and the whole process line were all less than 1, which meant that the production consumed a large amount of non-renewable resources, and the ELR was high. However, the ESDI of other steel enterprise were lower, being less than 0.1 [23,24]. The ESDI of the puddling and rolling processes were greater than 1 but less than 10, suggesting that the emergy sustainable development of puddling and rolling processes were in good condition. Considering all the useful outputs, such as intermediate and reusable products, emergy sustainability indexes (TESDI and NESDI) have been greatly improved. The TESDI and NESDI of puddling, rolling, and the whole process were within a reasonable range of 1 to 10. It could be seen that if the steel production line recycled the intermediate products of each process, it could achieve sustainable development; if not, the system has a high environmental load rate and cannot develop sustainably.

## 4. Conclusions and Recommendations

### 4.1. Conclusions

Based on the emergy of various input–output indicators, the total input and output emergy of the steel production line was not very different; the largest input was the intermediate products and recyclable materials produced in the production process; the recyclable materials accounted for 43% of the total input. The input emergy was mainly non-renewable resources, and the ELR was high; the emergy of pollutants discharged was very low, indicating that the environmental impact of steel production was small if the pollutants were discharged after treatment.

The ELR of pelletizing and sintering processes that occurs in the front-end production line was the highest; the proportions of recycled materials used for steel-making and puddling were the highest, and played the greatest role in 'waste' absorption. The EIR in rolling were the highest since its dependence on natural system was the greatest. The emergy value of pollutants from each process was very small, and the EnIR was close to or below 0.001. The PR was only 0.324 in the puddling process, and the emergy efficiency of production could greatly increase if the product rate of puddling was improved. The EYR of sintering and rolling processes were the highest. Both the TEYR and NEYR of puddling were the highest. There was little difference between the procedures in the EIOR, TEIOR, and NEIOR after considering all resource inputs simultaneously.

The ESDI of pelletizing, sintering and steel-making were less than 1, indicating an unsustainable production process but puddling and rolling processes were reasonable. Considering the intermediate products and recyclable materials, the TESDI and NESDI of puddling, rolling and the whole process were between 1 and 10, and the development was acceptable. Therefore, the steel production process could achieve sustainable development if various intermediate products could be recycled considerably.

### 4.2. Recommendations

This paper systematically analyzed the input and output of the steel production line, but the research process still needs to be improved and further explored. Pollutants discharged from the steel production process will have adverse effects on human and other biological health in the ecological environment. Due to absence of corresponding methods and data for assessing biological hazards, this part of the study was omitted for the time being. The pollutant could be evaluated more accurately once the biological hazards are considered in future studies. The type of pollutants from the steel production

process were much more varied than the particulate matter, sulfur dioxide, and nitrogen oxides studied here. After determining the influence of other pollutants for inclusion in future evaluations, the results would be more comprehensive.

In addition to emergy analysis, other eco-economic assessments have also been tried to evaluate the sustainability of steel production. For example, the life cycle assessment method, which mainly concerns the environmental impact of goods and services, has been used at different scales [31–34]. Although each method has its own advantages and disadvantages, it may be more scientific and informative to combine several eco-economic assessments with emergy analysis.

**Author Contributions:** Y.W. and F.M. designed this research; F.M. performed calculation and analyzed the data; and A.E.E. and F.M. wrote the paper. All authors have read and approved the final manuscript.

**Funding:** This research was funded by the Program of the Humanities and Social Sciences Research of Ministry of Education of China (17YJCZH192).

**Acknowledgments:** The authors wish to acknowledge the anonymous reviewers for their suggestions that have greatly improved our study.

**Conflicts of Interest:** The authors declare no conflict of interest.

# Appendix A

**Table A1.** Emergy input and output in the steel production sub-process.

| Production Process | System | Resource Type | Indicator | Unit | Raw Data | Emergy Conversion Rate sej/unit | Emergy sej |
|---|---|---|---|---|---|---|---|
| | **Input** | | | | | | |
| | Natural system | Renewable resources (R) | Fresh water | t | 60,000 | $6.64 \times 10^{11}$ [a] | $3.98 \times 10^{16}$ |
| | | Non-renewable resources (N) | Bentonite | t | 19,200 | $1.00 \times 10^{15}$ [a] | $1.92 \times 10^{19}$ |
| | | | Powdered iron | t | 1,212,000 | $8.55 \times 10^{14}$ [a] | $1.04 \times 10^{21}$ |
| | Human economics system | Non-renewable resources ($F_N$) | Electricity | J | $1.17 \times 10^{15}$ | 160,000 [b] | $1.87 \times 10^{20}$ |
| | Steel production system | Recycled materials | Blast furnace gas | J | $2.72 \times 10^{14}$ | 66,000 [a] | $1.79 \times 10^{19}$ |
| Pelletizing | **Output** | | | | | | |
| | Natural system | Pollutant | $SO_2$ | | | | $2.43 \times 10^{17}$ |
| | | | Dust | | | | $4.25 \times 10^{17}$ |
| | | | $NO_x$ | | | | $6.44 \times 10^{17}$ |
| | Human economics system | Products | Pellet | t | 1,200,000 | $1.09 \times 10^{15}$ [a] | |
| | | Recycled materials | Desulphurizing Slag | t | 6200 | $1.00 \times 10^{15}$ [a] | $6.20 \times 10^{18}$ |
| | | | Pellet dust removal ash | t | 48 | $8.30 \times 10^{14}$ [a] | $3.98 \times 10^{16}$ |
| | **Input** | | | | | | |
| | Natural system | Renewable resources (R) | Fresh water | t | 87,000 | $6.64 \times 10^{11}$ [a] | $5.78 \times 10^{16}$ |
| | | Non-renewable resources(N) | Domestic powdered iron | t | 1,625,000 | $8.55 \times 10^{14}$ [a] | $1.39 \times 10^{21}$ |
| | | | Powdered iron abroad | t | 6,025,000 | $8.55 \times 10^{14}$ [a] | $5.15 \times 10^{21}$ |
| | | | Limestone | t | 1,464,000 | $1.00 \times 10^{15}$ [a] | $1.46 \times 10^{21}$ |
| | | | High magnesium powder | t | 85,900 | $1.00 \times 10^{15}$ [a] | $8.59 \times 10^{19}$ |
| | Human economics system | Non-renewable resources ($F_N$) | Electricity | J | $1.17 \times 10^{15}$ | 160,000 [b] | $1.87 \times 10^{20}$ |
| | | | Coke powder | J | $1.11 \times 10^{16}$ | 10,600 [a] | $1.18 \times 10^{20}$ |
| | Steel production system | Recycled materials | Dust removal ash | t | 48,200 | $8.30 \times 10^{14}$ [a] | $4.00 \times 10^{19}$ |
| | | | Water treatment sludge | t | 53,850 | $1.00 \times 10^{15}$ [a] | $5.39 \times 10^{19}$ |
| | | | Sinter reentry | t | 184,400 | $1.09 \times 10^{15}$ [a] | $2.01 \times 10^{20}$ |
| | | | Pellet return | t | 100,300 | $1.09 \times 10^{15}$ [a] | $1.09 \times 10^{20}$ |
| | | | Blast furnace gas | J | $2.07 \times 10^{15}$ | 66,000 [a] | $1.37 \times 10^{20}$ |
| | | | Use of steam | J | $8.64 \times 10^{14}$ | 3090 [a] | $2.67 \times 10^{18}$ |
| Sintering | **Output** | | | | | | |
| | Natural system | Pollutant | $SO_2$ | | | | $1.90 \times 10^{18}$ |
| | | | Dust | | | | $4.81 \times 10^{18}$ |
| | | | $NO_x$ | | | | $5.46 \times 10^{18}$ |
| | Human economics system | Products | Sinter | t | 9,150,000 | $1.09 \times 10^{15}$ [a] | $9.98 \times 10^{21}$ |
| | | Recycled materials | Desulphurized gypsum | t | 23,600 | $1.00 \times 10^{15}$ [c] | $2.36 \times 10^{19}$ |
| | | | Desulphurization waste ash | t | 16,560 | $8.30 \times 10^{14}$ [a] | $1.38 \times 10^{19}$ |
| | | | Sinter reentry | t | 757,360 | $1.09 \times 10^{15}$ [a] | $8.26 \times 10^{20}$ |
| | | | Sintering dust | t | 75,900 | $8.30 \times 10^{14}$ [a] | $6.30 \times 10^{19}$ |
| | | | Steam generation | J | $6.66 \times 10^{14}$ | 3090 [a] | $2.06 \times 10^{18}$ |

**Table A1.** *Cont.*

| Production Process | System | Resource Type | Indicator | Unit | Raw Data | Emergy Conversion Rate sej/unit | Emergy sej |
|---|---|---|---|---|---|---|---|
| | **Input** | | | | | | |
| | Natural system | Renewable resources (R) | Compressed air | t | 1,217,695.5 | $5.16 \times 10^{13}$ [a] | $6.29 \times 10^{19}$ |
| | | | Fresh water | t | 1,260,000 | $6.64 \times 10^{11}$ [a] | $8.37 \times 10^{17}$ |
| | | Non-renewable resources (N) | Iron block | t | 101,700 | $8.55 \times 10^{14}$ [a] | $8.70 \times 10^{19}$ |
| | | | Coal | J | $5.09 \times 10^{15}$ | 39,801 [a] | $2.02 \times 10^{20}$ |
| | | | Pulverized coal | J | $6.59 \times 10^{14}$ | 39,801 [a] | $2.62 \times 10^{19}$ |
| | Human economics system | Non-renewable resources ($F_N$) | Electricity | J | $7.70 \times 10^{14}$ | 160,000 [b] | $1.23 \times 10^{20}$ |
| | | | Coke | J | $4.45 \times 10^{16}$ | 10,600 [a] | $4.71 \times 10^{20}$ |
| | | | Nut coke | J | $7.32 \times 10^{14}$ | 10,600 [a] | $7.76 \times 10^{18}$ |
| | Steel production system | Intermediate products | Sinter | t | 6,630,000 | $1.09 \times 10^{15}$ [a] | $7.23 \times 10^{21}$ |
| | | | Pellet | t | 1,200,000 | $1.09 \times 10^{15}$ [a] | $1.31 \times 10^{21}$ |
| Puddling | | Recycled materials | Use of steam | J | $1.73 \times 10^{14}$ | 3090 [a] | $5.34 \times 10^{17}$ |
| | | | Blast furnace gas | J | $1.43 \times 10^{16}$ | 66,000 [a] | $9.43 \times 10^{20}$ |
| | **Output** | | | | | | |
| | Natural system | Pollutant | $SO_2$ | | | | $1.47 \times 10^{17}$ |
| | | | Dust | | | | $3.55 \times 10^{18}$ |
| | | | $NO_x$ | | | | $5.44 \times 10^{17}$ |
| | Human economics system | Products | Pig iron | t | 4,650,000 | $2.71 \times 10^{15}$ [a] | $1.26 \times 10^{22}$ |
| | | Recycled materials | Blast furnace slag | t | 1,417,000 | $8.55 \times 10^{14}$ [a] | $1.21 \times 10^{21}$ |
| | | | Gas ash | t | 57,500 | $8.30 \times 10^{14}$ [a] | $4.77 \times 10^{19}$ |
| | | | Blast furnace gas | J | $1.79 \times 10^{16}$ | 66,000 [a] | $2.12 \times 10^{21}$ |
| | | | Hot blast stove flue gas | t | 27,619,708.4 | $8.30 \times 10^{14}$ [a] | $2.29 \times 10^{22}$ |
| | | | Dust and ash | t | 39,100 | $8.30 \times 10^{14}$ [a] | $3.25 \times 10^{19}$ |
| | **Input** | | | | | | |
| | Natural system | Renewable resources (R) | Fresh water | t | 1,176,000 | $6.64 \times 10^{11}$ [a] | $7.81 \times 10^{17}$ |
| | | | Nitrogen consumption | t | 4,515,739.44 | $2.00 \times 10^{14}$ [d] | $9.03 \times 10^{20}$ |
| | | | Oxygen consumption | t | 2,667,180.15 | $5.16 \times 10^{13}$ [b] | $1.38 \times 10^{20}$ |
| | | | Compressed air | t | 1,178,415 | $5.16 \times 10^{13}$ | $6.08 \times 10^{19}$ |
| | | Non-renewable resources (N) | Iron block | t | 110,000 | $8.55 \times 10^{14}$ | $9.41 \times 10^{19}$ |
| Steel-making | | | Ferroalloy | t | 13,300 | $8.55 \times 10^{14}$ | $1.14 \times 10^{19}$ |
| | | | Doomite | t | 65,500 | $1.00 \times 10^{15}$ [a] | $6.55 \times 10^{19}$ |
| | | | Flour | t | 600 | $1.00 \times 10^{15}$ [a] | $6.00 \times 10^{17}$ |
| | Human economics system | Non-renewable resources ($F_N$) | Electricity | J | $7.85 \times 10^{14}$ | 160,000 [b] | $1.26 \times 10^{20}$ |
| | | | White ash | t | 468,000 | $2.16 \times 10^{15}$ [a] | $1.01 \times 10^{21}$ |
| | Steel production system | Intermediate products | Pig iron | t | 4,650,000 | $2.17 \times 10^{15}$ [a] | $1.01 \times 10^{22}$ |
| | | Recycled materials | Steel scrap | t | 32,000 | $3.09 \times 10^{15}$ [a] | $9.89 \times 10^{19}$ |
| | | | Blast furnace gas | J | $2.86 \times 10^{14}$ | 66,000 [a] | $1.89 \times 10^{19}$ |
| | | | Convertor gas | J | $1.29 \times 10^{15}$ | 66,000 [a] | $8.52 \times 10^{19}$ |

**Table A1.** *Cont.*

| Production Process | System | Resource Type | Indicator | Unit | Raw Data | Emergy Conversion Rate sej/unit | Emergy sej |
|---|---|---|---|---|---|---|---|
| Steel-making | **Output** | | | | | | |
| | Natural system | Pollutant | Dust | | | | $2.63 \times 10^{18}$ |
| | Human economics system | Products | Billet steel | t | 4,500,000 | $3.09 \times 10^{15}$ [a] | |
| | | Recycled materials | Steel slag | t | 427,600 | $3.09 \times 10^{15}$ [a] | $1.32 \times 10^{21}$ |
| | | | Dust mud | t | 53,800 | $1.00 \times 10^{15}$ [a] | $5.38 \times 10^{19}$ |
| | | | Dust | t | 120 | $8.30 \times 10^{14}$ [a] | $9.96 \times 10^{16}$ |
| | | | Iron oxide skin | t | 27,800 | $8.55 \times 10^{14}$ [a] | $2.38 \times 10^{19}$ |
| | | | Remainder residue | t | 19,500 | $8.30 \times 10^{14}$ [a] | $1.62 \times 10^{19}$ |
| | | | Steam generation | J | $1.01 \times 10^{15}$ | 3090 [a] | $3.13 \times 10^{18}$ |
| | | | Convertor gas | J | $4.35 \times 10^{15}$ | 66,000 [a] | $2.87 \times 10^{20}$ |
| Rolling | **Input** | | | | | | |
| | Natural system | Renewable resources (R) | compressed air | t | 104,472.48 | $5.16 \times 10^{13}$ [a] | $5.39 \times 10^{18}$ |
| | | | Fresh water | t | 670,000 | $6.64 \times 10^{11}$ [a] | $4.45 \times 10^{17}$ |
| | Human economics system | Non-renewable resources ($F_N$) | Electricity | J | $1.44 \times 10^{15}$ | 160,000 [b] | $2.30 \times 10^{20}$ |
| | Steel production system | Intermediate products | Billet steel | t | 3,243,700 | $3.09 \times 10^{15}$ [a] | $1.00 \times 10^{22}$ |
| | | Recycled materials | Blast furnace gas | J | $3.90 \times 10^{15}$ | 66,000 [a] | $2.58 \times 10^{20}$ |
| | **Output** | | | | | | |
| | Natural system | Pollutant | $SO_2$ | | | | $6.95 \times 10^{16}$ |
| | | | Dust | | | | $1.88 \times 10^{17}$ |
| | | | $NO_x$ | | | | $2.07 \times 10^{17}$ |
| | Human economics system | Products | Rolled steel | t | 3,200,000 | $3.09 \times 10^{15}$ [a] | |
| | | Recycled materials | Iron oxide sludge | t | 21,900 | $1.00 \times 10^{15}$ [a] | $2.19 \times 10^{19}$ |
| | | | Steel scrap | t | 28,300 | $3.09 \times 10^{15}$ [a] | $8.74 \times 10^{19}$ |
| | | | Steam generation | J | $3.95 \times 10^{14}$ | 3090 [a] | $1.22 \times 10^{18}$ |
| Oxygen production | **Input** | | | | | | |
| | Natural system | Renewable resources (R) | air | t | 1,964,117.14 | $5.16 \times 10^{13}$ [a] | $1.01 \times 10^{20}$ |
| | | | Fresh water | t | 286,651.81 | $6.64 \times 10^{11}$ [a] | $1.90 \times 10^{17}$ |
| | Human economics system | Non-renewable resources ($F_N$) | Electricity | J | $7.93 \times 10^{14}$ | 160,000 [b] | $1.27 \times 10^{20}$ |
| Power plant | **Input** | | | | | | |
| | Steel production system | Recycled materials | Blast furnace gas | J | $1.14 \times 10^{16}$ | 65,999 [a] | $7.50 \times 10^{20}$ |
| | | | Convertor gas | J | $3.06 \times 10^{15}$ | 66,000 [a] | $2.02 \times 10^{20}$ |
| | | | Steam generation | J | $1.67 \times 10^{16}$ | 3090 [a] | $5.15 \times 10^{19}$ |
| | | | Use of steam | J | $1.67 \times 10^{16}$ | 3090 [a] | $5.15 \times 10^{19}$ |
| | | | Electricity | J | $1.89 \times 10^{15}$ | 160,000 [b] | $3.03 \times 10^{20}$ |
| | **Output** | | | | | | |
| | Natural system | Pollutant | $SO_2$ | | | | $4.42 \times 10^{17}$ |
| | | | Dust | | | | $4.34 \times 10^{17}$ |
| | | | $NO_x$ | | | | $5.15 \times 10^{17}$ |

**Table A1.** *Cont.*

| Production Process | System | Resource Type | Indicator | Unit | Raw Data | Emergy Conversion Rate sej/unit | Emergy sej |
|---|---|---|---|---|---|---|---|
| | **Input** | | | | | | |
| Whole process | Human economics system | Renewable resources (R) | Labor | 人 | 8500 | $3.10 \times 10^{16}$ a | $2.64 \times 10^{20}$ |
| | | | Investment in fixed assets | $ | 46,937,658.6 | $4.94 \times 10^{12}$ e | $2.32 \times 10^{20}$ |
| | | | Soil loss | g | $4.43 \times 10^{11}$ | $1.70 \times 10^{9}$ c | $7.53 \times 10^{20}$ |

Note: ① The calculation of the emergy of pollutants is detailed in the Section 3.2.2; ② Conversion parameters of some raw data: Compressed air mass: volume × 1.239 g/L; steam heat: mass × 2817.2381 J/g; oxygen mass: volume × 10,470 g/m$^3$; nitrogen mass: volume × 9168.8 g/m$^3$; coke heat: mass × 28,470 J/g; coal heat: mass × 8374 J/g; electric heat: kWh × 3.6 × 10$^6$ J/kWh; blast furnace gas heat: volume × 3344 kJ/m$^3$; Convertor gas heat: volume × 7527 kJ/m$^3$; Lifetime of factory: 20 years. ③ Energy of soil loss = soil loss mass organic matter content × soil organic matter calorific value; soil organic matter calorific value is 106 kcal/t; surface soil thickness is 0.15 m; organic matter content is 5%; soil bulk density is 1.3 g/cm$^3$. ④ Labor, fixed assets investment and soil loss are only counted in the total production process of factory for the three indicators are not easy to collected in subsystems. ⑤ Reference: a: [35]; b: [23]; c: [36]; d: [37]; e: [38].

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
