# Peer review of "An Evaluation of Input–Output Value for Sustainability in a Chinese Steel Production System Based on Emergy Analysis"

_sustainability, doi:10.3390/su10124749_

Round 1

Reviewer 1 Report

the paper is well structured and has scientific soundness supported by clear description of the methodology, the underpinning data and the results.

I would only suggest to make sure that the tables are either on a single page or with headings repeated at the top of each page. alternatively use charts.

do not use sub-sections in the conclusions

Author Response

Comments:

The paper is well structured and has scientific soundness supported by clear description of the methodology, the underpinning data and the results.

1) I would only suggest to make sure that the tables are either on a single page or with headings repeated at the top of each page. alternatively use charts.do not use sub-sections in the conclusions

ReplyRevised accordingly (lines 186,354-356 in the revised manuscript).

2) do not use sub-sections in the conclusions

ReplyRevised accordingly (lines 315,321,330 in the revised manuscript).

Reviewer 2 Report

The topic is interesting but I have major remarks.

Concerning the title and how you generalized the results of your article , you have studied only one case thus  not use "the steel production system" but a steel production system or a Chinese steel production...

Introduction: 

L31: reference fort the value of the global turnover and at this moment?

L37: social system resources: unclear for me, give more explanations because this vocabulary seems related to sociology area and not usual for other sciences.

L50: Specify reference for Odum

L51-52: Solar energy value not enough detailed: Solar Emjoule can be defined here.

L55-59: This sentence is not clear for me because the unit of energy is the Joule and this why we can compare all forms of energy thus your sentence is not appropriate. You have to give a clear presentation of emergy theory and particularly the notion of transformity.

Methodology: I have major remarks concerning this paragraph.

Concerning the resources classification: Why you don't use R N F as made in standardized emergy analysis, to class the resources it's more easy to separate for example renewable or green electricity from other electricity coming from coal. I am not convinced that electricity can be classified into the human economic system.

You have to define the limits of the system and your diagram is not a standardized flow diagram and without legend. 

For data collection give reference for the report used.  How you have the accessibility to these data, not clear for me?

L117 What is quality data?

A major criticism concerns calculation of emergy for emissions. This point should be more discussed. You have used the method of Ulgiati and Brown, 2002 (please indicates this reference but  I am not convinced by this approach to calculate the emergy in relationships only with the dilution of pollutants. Give more argues to validate this approach. Pollutants are losses of the system or considered as by-products and your formulae are correct to calculate the emergy of a source as for windpower for example, otherwise there will be an overestimation of emergy value for pollutants considered as a source for recycling processes for example. I think that this is a speculative point of view. In my point of view emergy determination is correct if you take into account the historical embodied energy through the transformity coefficient and produced exergy. Here you calculate emergy exported out of the system through a possible impact, this should be argue. the assumptions and postulates of this method (Ulgiati and Brown) are criticizable. 

results and discussion

I think that quantifying ecological and economic losses applying emergy analysis should be more discussed. Why any comments with results obtained from other studies on steel production in China (references such as 18, 19 or 20)? 

What additional information brings your study, where is the novelty? It's not clear for me.

Author Response

Comments:

The topic is interesting but I have major remarks.

1) Concerning the title and how you generalized the results of your article , you have studied only one case thus not use "the steel production system" but a steel production system or a Chinese steel production...

ReplyWe have added ' A Chinese' in the title (see line 3 in the revised manuscript)

2) Introduction: 

L31: reference fort the value of the global turnover and at this moment?

ReplyWe have revised that sentence as “and the world’s crude steel output exceeded 1.6 billion tons in 2016”(see line 32 in the revised manuscript )

L37: social system resources: unclear for me, give more explanations because this vocabulary seems related to sociology area and not usual for other sciences.

ReplyWe have revised to 'human economic system' in throughout the paper (see lines 38,39;116; 129;130;133,etc. in the revised manuscript)

L50: Specify reference for Odum

ReplyWe have added the reference (see lines 49-51 in the revised manuscript)

L51-52: Solar energy value not enough detailed: Solar Emjoule can be defined here.

ReplyRevised accordingly (line 55 in the revised manuscript).

L55-59: This sentence is not clear for me because the unit of energy is the Joule and this why we can compare all forms of energy thus your sentence is not appropriate. You have to give a clear presentation of emergy theory and particularly the notion of transformity.

ReplyWe have explained the emergy theory and transformity again (see lines 49-68 in the revised manuscript)

3) Methodology: I have major remarks concerning this paragraph.

Concerning the resources classification: Why you don't use R N F as made in standardized emergy analysis, to class the resources it's more easy to separate for example renewable or green electricity from other electricity coming from coal. I am not convinced that electricity can be classified into the human economic system.

ReplyWe have used R N F Y according to the standardized emergy analysis (see line 119-122;Figure 2; Table 2 and Schedule 1 in the revised manuscript). However, we  classified electricity under F, as in the following previous reports: ([1]Lu, H. F.; Kang, W. L.; Campbell, D. E.; Ren, H.; Tan, Y. W.; Feng, R. X.; Luo, J. T.; Chen, F. P., Emergy and economic evaluations of four fruit production systems on reclaimed wetlands surrounding the Pearl River Estuary, China. Ecological Engineering 2009, 35, (12), 1743-1757. 

[2] Li, Q.; Yan, J. M., Assessing the health of agricultural land with emergy analysis and fuzzy logic in the major grain-producing region. Catena 2012, 99, 9-17.

[3] La Rosa, A. D.; Siracusa, G.; Cavallaro, R., Emergy evaluation of Sicilian red orange production. A comparison between organic and conventional farming. J Clean Prod 2008, 16, (17), 1907-1914. 

[4] Ciotola, R. J.; Lansing, S.; Martin, J. F., Emergy analysis of biogas production and electricity generation from small-scale agricultural digesters. Ecological Engineering 2011, 37, (11), 1681-1691. )

You have to define the limits of the system and your diagram is not a standardized flow diagram and without legend. 

ReplyWe have defined the boundary of the system (see line 111-112 in the revised manuscript) and the diagram has been revised according to the standardized flow diagram with legends added (see Figure 2 in the revised manuscript).

For data collection give reference for the report used.  How you have the accessibility to these data, not clear for me?

ReplyWe have clarified the report again (see lines 140-142 in the revised manuscript).

L117 What is quality data?

ReplyWe have revised to ' mass' (see line 149 in the revised manuscript)

A major criticism concerns calculation of emergy for emissions. This point should be more discussed. You have used the method of Ulgiati and Brown, 2002 (please indicates this reference but  I am not convinced by this approach to calculate the emergy in relationships only with the dilution of pollutants. Give more argues to validate this approach. Pollutants are losses of the system or considered as by-products and your formulae are correct to calculate the emergy of a source as for windpower for example, otherwise there will be an overestimation of emergy value for pollutants considered as a source for recycling processes for example. I think that this is a speculative point of view. In my point of view emergy determination is correct if you take into account the historical embodied energy through the transformity coefficient and produced exergy. Here you calculate emergy exported out of the system through a possible impact, this should be argue. the assumptions and postulates of this method (Ulgiati and Brown) are criticizable. 

ReplyThe point is well taken; although we referenced the method of Ulgiati and Brown (2002), our method was not entirely the same as theirs (see lines 167-173 in the revised manuscript).

4)results and discussion

I think that quantifying ecological and economic losses applying emergy analysis should be more discussed. Why any comments with results obtained from other studies on steel production in China (references such as 18, 19 or 20)?

ReplyThe discussion on quantifying loss of pollutants is now added in lines 169-175 in the revised manuscript with some comments (see lines 209-210; 218-221; 235-236; 249-250; 256-258;273-274;305 in the revised manuscript).

What additional information brings your study, where is the novelty? It's not clear for me.

ReplyWe have added more information about the innovativeness (see lines 93-94 in the revised manuscript).

Reviewer 3 Report

This study is interestings and well-ogrnaized, and I have two small suggestions:

A comprehensive literature review of emergy synthesis on sustainability assessment or indutrial system is prerequisite, especially thoese focusing on renewable energies such as biodiese and biofuel.

2. Comparsions with soem other methods.

Author Response

Comments:

This study is interestings and well-ogrnaized, and I have two small suggestions:

1) A comprehensive literature review of emergy synthesis on sustainability assessment or indutrial system is prerequisite, especially thoese focusing on renewable energies such as biodiese and biofuel.

ReplyWe have added more literature review (see lines 73-80 in the revised manuscript).

2) Comparsions with soem other methods.

ReplyWe have added some comparative comments about other methods (see lines 346-350 in the revised manuscript).

Reviewer 4 Report

This manuscript has several major insufficient points as follows.

Structure of the manuscript:

There is not a clear explanation about the aim of this study. I guess the paragraph between lines 69 to 80 seems to explain that. But, the description is just writing what you have done in this study.

I could not understand the reason why Chapter 2 is independent from other chapters, which is too short and which seems to be a part of data collection.

Terminology and definitions:

As far as I read, I could not find any definition of several terms used in this manuscript, such as social system, natural system, steel production system, the natural ecosystem, human economic system and so on. I would say that authors shall describe the reason why those classification is required before definitions.

Some terms are used not in a general usage, which leads to less readability of this manuscript.

Some technical terms are not correct. The term ‘fire-proof materials’ is used for materials used on buildings. In iron and steel industry, they use refractory. The term ‘sintering machine’ might be a sintering furnace. Molten iron might be Pig iron. Conticaster seems not to be a technical term. Is ‘Continuously cast’ a name of materials provided from a conticaster? Heating furnace may be Annealing furnace. I have never heard a ligation machine in iron and steel industry. In this sequence of processes, an acid pickling process is missing? For describing iron and steel making processes, please refer technical papers or books written in English.

Author Response

Dear Editor,

Thanks very much for forwarding comments of reviewer No. 4 for inclusion in our revision. We have now made changes based on his/her comments, in addition to earlier revisions based on comments of the first three reviewers. The changes are marked in blue font color and are highlighted below:

Point 1: Structure of the manuscript:

There is not a clear explanation about the aim of this study. I guess the paragraph between lines 69 to 80 seems to explain that. But, the description is just writing what you have done in this study.

Response 1: We have added more information on the aim of this study (see lines 86-88 in the revised manuscript).

Point 2: I could not understand the reason why Chapter 2 is independent from other chapters, which is too short and which seems to be a part of data collection.

Response 2: We have revised Chapter 2 as a part of data collection (see lines 135-150 in the revised manuscript).

Point 3: Terminology and definitions:

As far as I read, I could not find any definition of several terms used in this manuscript, such as social system, natural system, steel production system, the natural ecosystem, human economic system and so on. I would say that authors shall describe the reason why those classification is required before definitions.

Some terms are used not in a general usage, which leads to less readability of this manuscript.

Response 3: We have added more information clarifying the meaning and usefulness of the  several terms and described the reasons for classification (see lines 106-111 in the revised manuscript).

Point 4: Some technical terms are not correct. The term fire-proof materialsis used for materials used on buildings. In iron and steel industry, they use refractory. The term sintering machinemight be a sintering furnace. Molten iron might be Pig iron. Conticaster seems not to be a technical term. Is Continuously casta name of materials provided from a conticaster? Heating furnace may be Annealing furnace. I have never heard a ligation machine in iron and steel industry. In this sequence of processes, an acid pickling process is missing? For describing iron and steel making processes, please refer technical papers or books written in English.

Response 4: We have revised accordingly (see Figure 2. in the revised manuscript).

Round 2

Reviewer 2 Report

Effectively the manuscript has been significantly improved becoming now acceptable for publication.